# A New Drought Index for Soil Moisture Monitoring Based on MPDI-NDVI Trapezoid Space Using MODIS Data

**Liangliang Tao [1,2], Dongryeol Ryu [2,*], Andrew Western [2] and Dale Boyd [3]**

1. Collaborative Innovation Center on Forecast and Evaluation of Meteorological Disaster, School of Geographical Sciences, Nanjing University of Information Science & Technology, Nanjing 210044, China; taoll09@nuist.edu.cn
2. Department of Infrastructure Engineering, University of Melbourne, Parkville, VIC 3010, Australia; a.western@unimelb.edu.au
3. Biosecurity and Agriculture Services Branch, Department of Jobs, Precincts and Regions, Echuca, VIC 3564, Australia; dale.boyd@agriculture.vic.gov.au
* Correspondence: dryu@unimelb.edu.au; Tel.: +61-403257335

**Abstract:** The temperature vegetation dryness index (TVDI) has been commonly implemented to estimate regional soil moisture in arid and semi-arid regions. However, the parameterization of the dry edge in the TVDI model is performed with a constraint to define the maximum water stress conditions. Mismatch of the spatial scale between visible and thermal bands retrieved from remotely sensed data and terrain variations also affect the effectiveness of the TVDI. Therefore, this study proposed a new drought index named the condition vegetation drought index (CVDI) to monitor the temporal and spatial variations of soil moisture status by substituting the land surface temperature (LST) with the modified perpendicular drought index (MPDI). In situ soil moisture observations at crop and pasture sites in Victoria were used to validate the effectiveness of the CVDI. The results indicate that the dry and wet edges in the parameterization scheme of the CVDI formed a better-defined trapezoid shape than that of the TVDI. Compared with the MPDI and TVDI for soil moisture monitoring at crop sites, the CVDI exhibited a performance superior to the MPDI and TVDI in most days where the coefficients of determination ($R^2$) achieved can reach to 0.67 on DOY023, 137, 274 and 0.71 on DOY 322 and reproduced more accurate spatial and seasonal variation of soil moisture. Moreover, the CVDI showed higher correlation with the Australian Water Resource Assessment Landscape (AWRA-L) soil moisture product on temporal scales. The $R^2$ can reach to 0.69 and the root mean square error (RMSE) is also much better than that of the MPDI and TVDI. Overall, it can be concluded that the CVDI appears to be a feasible method and can be successfully used in regional soil moisture monitoring.

**Keywords:** soil moisture; TVDI; condition vegetation drought index; AWRA-L; MPDI; MODIS

## 1. Introduction

Spatio-temporal distribution and variation of soil moisture (SM) serves as a critical parameter relative to the energy exchange in the soil–vegetation–atmosphere ecosystem, as it controls a wide range of hydrologic, meteorological and agricultural processes [1]. It determines the partitioning of precipitation into surface infiltration, runoff, evapotranspiration and subsoil drainage [2,3]. Accurate monitoring and assessment of SM is of crucial importance in arid areas where water deficit is gradually becoming the limited factor restricting agricultural productivity and ecological development. Australia, one of the driest countries on the earth, has experienced several major droughts and droughts have become more recurrent in the last 50 years, particularly in the south-eastern part of Australia [4]. It is predicted that Australia will become even drier due to decreasing rainfall and increasing temperature over the coming decades [5]. Therefore, estimating the

spatial and temporal dynamics of SM on regional and global scales is essential for guiding drought prediction and cropland irrigation scientifically.

Satellite remote sensing is typically regarded as the only viable technique for the regional and global monitoring of SM in a spatially and temporally feasible manner. It can provide land surface parameters related to SM continuously in a wide spectrum ranging from visible to microwave bands. Microwave-based remote sensing is commonly recognized as a promising tool to measure land surface SM regardless of weather and sunlight conditions [6]. However, the active microwave sensors have small repeat intervals (about 6–25 days) with a severe interference by surface vegetation, and the spatial resolution of passive microwave sensors (20–40 km) is too coarse for the majority of local applications [7–9].

Optical remote sensing methods have been developed using surface temperature, albedo and vegetation indices retrieved from visible, near infrared, short-wave infrared and thermal infrared data for SM modelling. Over the past 50 years, numerous retrievals based on these remotely sensed observations have been carried out to map SM on various spatial scales. These models mainly consist of reflectance-based methods, thermal infrared-based models and synergistic methods of visible with thermal infrared observations [10]. In the reflectance-based methods, great progress has been made to identify high correlation between annual or monthly integrated vegetation indices (VI) (such as normalized difference vegetation index (NDVI), normalized difference water index (NDWI), and enhanced vegetation index (EVI)) and drought-related factors (such as SM, precipitation) under a partial or full range of vegetation cover [11,12]. Meanwhile, many NDVI-based indices are often developed and widely applied in SM and drought monitoring such as the anomaly vegetation index (AVI), vegetation condition index (VCI) and vegetation anomaly index (VAI) [13–15]. However, these indices utilizing only reflected spectral information are strongly affected by soil texture, surface roughness, organic matters, and plant cover. For thermal infrared-based models, land surface temperature (LST) is significantly sensitive to SM by effectively taking advantage of the energy balance principle and water circulation under bare soil or sparse vegetation-covered conditions. However, the relationship between land surface temperature and SM presents poorly over highly vegetated surfaces. Consequently, the synergistic use of remotely sensed data acquired simultaneously from visible and thermal infrared bands is conducted to derive regional estimates of SM, for example, the available surface temperature-vegetation index (LST-VI) triangle or trapezoid method [16–18]. The slope of LST-VI space is generally regarded as an indicator of the "wetness" of land surface, which is physically characterized by the variations of soil water evaporation and vegetation transpiration.

For directly estimating SM by linking the physical properties of LST-VI space with an index, Sandholt et al. [19] proposed the temperature vegetation dryness index (TVDI) comprehensively considering the influence of LST and VI on SM estimates on large spatial scales. This index can be applied in a region that has mixed land covers, and only satellite-derived information is required [20,21]. The spatio-temporal variations of SM can be indirectly assessed by use of an empirical relationship between TVDI and SM. Son et al. [22] utilized monthly NDVI and LST data to compare the efficiency of TVDI with the crop water stress index (CWSI), and the results indicated that the TVDI yielded close correlations with CWSI but was more sensitive to SM stress than CWSI. Holzman et al. [23] estimated regional crop yield using TVDI based on LST and EVI space, and the results showed that TVDI had a strong correlation with SM and it also was in agreement with the spatial pattern of SM. Chen et al. [24] and Shi et al. [25] both explored the applicability of TVDI to assess its sensitivity to different land-cover types. They found that there is a highly negative correlation between TVDI and SM data, and the TVDI can reflect the SM status in most types of land cover.

Several assumptions and limitations need to be considered when applying the TVDI: (1) a given spatial domain is needed to present SM and vegetation cover conditions, and the dimension of the mapping area should be large enough to ensure that the scatterplots

of LST versus VI can adequately form a regular triangular or trapezoid shape including completely wet edges and extremely dry edges; (2) in the spatial domain, surface properties and atmospheric conditions are relatively homogeneous and the variability of SM simply depends on variations of LST; (3) land cover types and topography should be fully considered to reduce the uncertainty of TVDI in SM retrieval. Previous studies have proposed many calibrated or improved TVDI methods according to these assumptions and limitations, which are classified into three categories: (1) a temperature correction model using surface–air temperature differences based on regional DEM data [26–29], day–night temperature differences [30] to effectively eliminate the impact of solar radiation and atmospheric background reflectance; (2) a dry edge correction method [31] or bi-parabolic LST-NDVI space method [32,33] to reshape the fitting equations of the dry edge or wet edge; (3) a time domain solution considering the maximum surface temperature of bare soil, which is a special case of TVDI [7,34]. These improvements have considerable scientific significances and provide promising approaches with potential application for SM estimation and drought monitoring.

Additionally, it should be noted that, for LST-VI space, most satellites hardly provide the same spatial resolution of visible, infrared and thermal bands data [35]. Spatial and spectral samplings will result in loss of some useful information and pose uncertainties in SM estimates. Meanwhile, sufficient meteorological parameters including air temperature, solar radiation, wind velocity, atmospheric emissivity, etc., are needed in many studies, such as determination of the theoretical dry edge and wet edge [36] and LST retrieval from Landsat data [37]. Ghulam et al. [38] proposed a vegetation condition albedo drought index (VCADI) based on an albedo-NDVI triangular space to substitute LST with surface broadband albedo in the near infrared band. Lu et al. [39] used apparent thermal inertia (ATI) instead of LST to develop an ATI-NDVI space for SM monitoring. Wang et al. [40] developed a triangular space formed by perpendicular drought index (PDI) and NDVI to estimate evapotranspiration (ET). However, PDI poorly performed under densely vegetated conditions and non-flat topography with different soil types. Compared with PDI, the modified PDI (MPDI) proposed by Ghulam et al. [41] was relatively highly accurate and more sensitive to SM in areas with vegetation cover.

In this study, a new drought index is developed by combing the MPDI with the parameterization scheme of TVDI. LST in the LST-VI space is substituted with the MPDI to construct a new feature space with quite similar shape and spectral patterns. The modification is meant to improve the performance of the drought index in SM and drought monitoring. In order to evaluate the effectiveness of the new drought index, a comparison of SM estimation between the new drought index, TVDI and MPDI is constructed based on SM measurements in local stations and remotely sensed data. Then, the Australian Water Resource Assessment Landscape (AWRA-L) SM product is used to further validate the effectiveness of the drought index. Finally, the effects of land cover types and data used in this study on the drought indices are discussed, then followed by the conclusion.

## 2. Study Area and Data

### 2.1. Study Area

Victoria is a state of Australia situated in south-eastern part of the country within the latitude of 33°59′~39°12′S and longitude of 140°58′~149°59′E, which covers an area of 237,659 km$^2$ and is generally high lands having an altitude of 1000~2000 m (Figure 1). It has a characteristic of Mediterranean climate with hot, dry summers and cool, wet winters. Its climate varies from cool along the coast to semi-arid temperate with hot summers in the north-west. A mountain and cooler climate is produced in the center of the state, while the coastal plain south of state has a wetter and wilder climate than Victoria's other areas. In the northwest of Victoria, hot winds blow from nearby semi-deserts with average temperatures of 32 °C during summer and 15 °C in winter. The Victorian Alps in the northwest have average temperatures of lower than 9 °C in winter and even below 0 °C in the highest regions, which is the coldest part of Victoria. Rainfall in Victoria is mainly

concentrated in the mountains and coastal areas of the northeast, and even in these parts of the country, the annual rainfall exceeds 1800 mm.

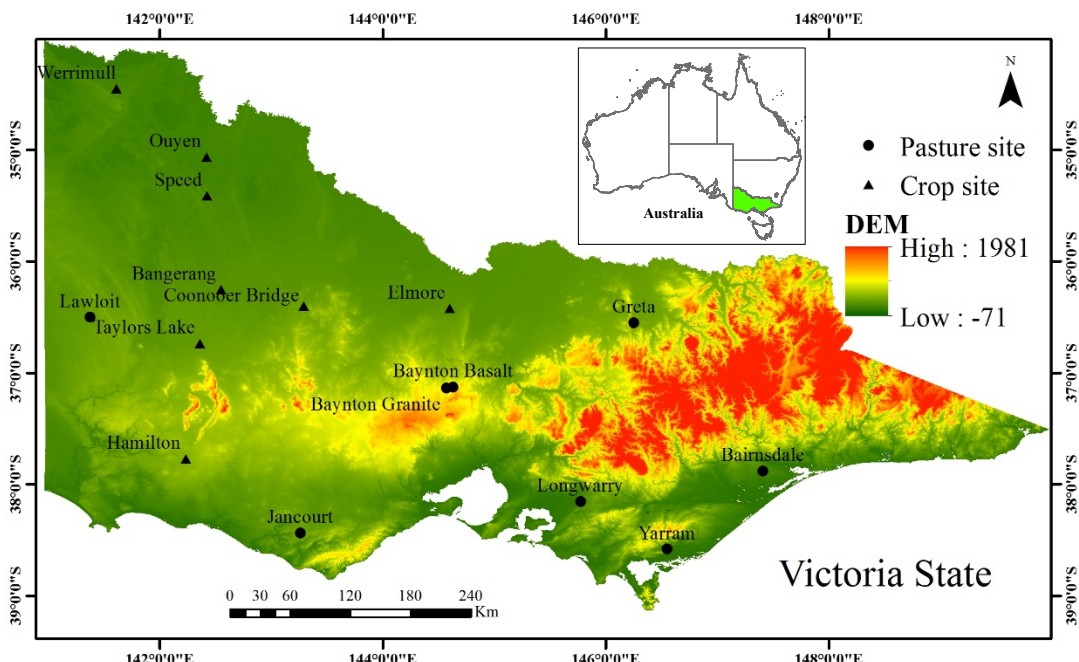

**Figure 1.** Distribution of crop and pasture sites over the local stations and the geographic location of the study area.

Land cover in Victoria is classified into urban area, bare land, water bodies and vegetated cover such as woody vegetation, pasture/grassland and cereal crops (Figure 2). The woody vegetation includes native forests, remnant vegetation, hardwood and softwood plantations, which has the largest proportion amounting to 42.22% of the state, according to the Victorian Land Use Information System (VLUIS). The pasture/grassland describes any herbaceous ground cover that is present for most of the year and totals 31.15% of the state. The dominant crop is grain predominantly located in western and northern Victoria and the growing of grain is continuing to expand into the high rainfall zones of southern Victoria. The grain production accounts for approximately 11% of Australia's production which has fluctuated over recent years primarily due to weather events.

### 2.2. Remote Sensing Data

Considering the constraints of the study area being large and covering a wide area, the Moderate Resolution Imaging Spectroradiometer (MODIS) products with temporal and spatial characteristics of different land cover types were selected to retrieve time series of representative of the spatial variation in water stress and vegetation status. Specifically, daily atmospheric corrected reflectance product at 500 m resolution (MOD09GA) and daily land surface temperature product at 1 km resolution (MOD11A1) were used to calculate drought indices and verify the estimated model results with measured volumetric SM data, which were both downloaded from NASA's official website (https://ladsweb.modaps.eosdis.nasa.gov/search/). All MODIS products have been geometrically and radiometrically corrected and were originally stored in a sinusoidal projection system. So, the MODIS Reprojection Tool was used to reproject these images and unify the spatial resolution to 1 km. In order to effectively reduce the influence of cloud on the MODIS products, cloud-free and daytime images for 18 days in the year 2019 were selected in this study. Table 1 presents the image information including day of year (DOY), date, overpass time of satellite, the minimum and maximum of LST.

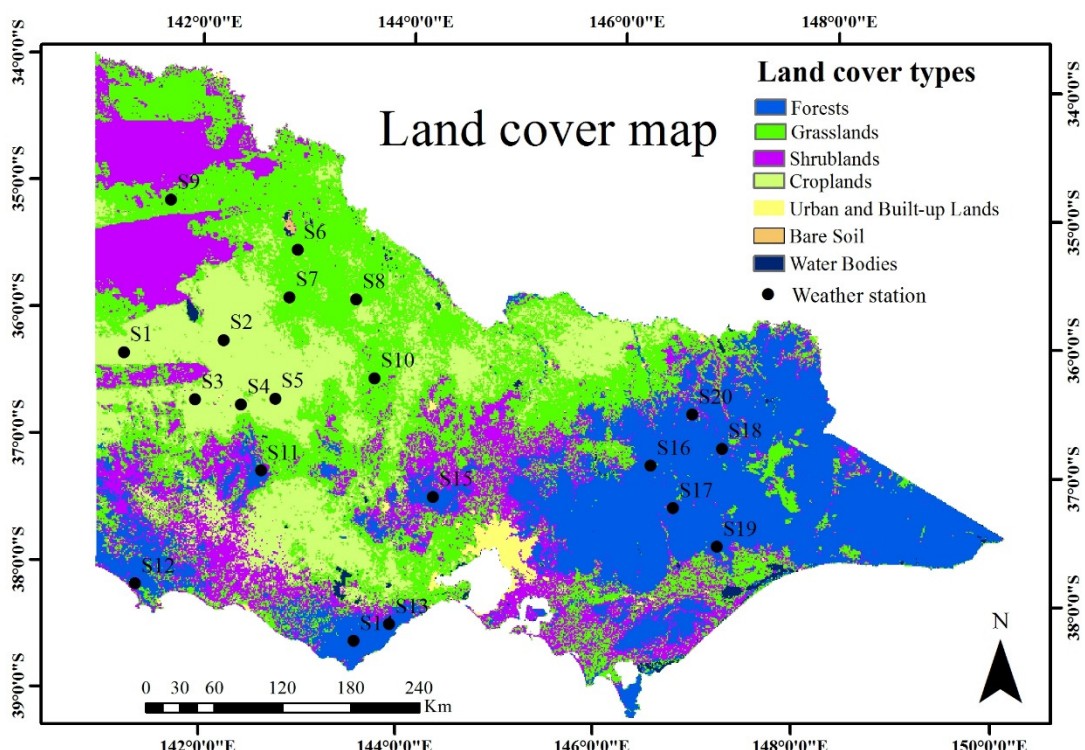

**Figure 2.** Land cover map of the study area.

**Table 1.** Details of Moderate Resolution Imaging Spectroradiometer (MODIS) images for 18 days in 2019.

| DOY | Date | Overpass Time (UTC) | LST (K) |
|:---:|:---:|:---:|:---:|
| 004 | 4 January | 00:42 | [295.4, 333.3] [1] |
| 023 | 23 January | 01:14 | [287.6, 331.8] |
| 055 | 24 February | 01:17 | [290.6, 322.2] |
| 061 | 2 March | 00:42 | [291.9, 330.9] |
| 075 | 16 March | 00:55 | [277.8, 317.6] |
| 102 | 12 April | 01:18 | [280.8, 307.4] |
| 137 | 17 May | 01:55 | [276.7, 297.7] |
| 162 | 11 June | 01:50 | [276.2, 294.7] |
| 176 | 25 June | 00:22 | [267.4, 294.0] |
| 178 | 27 June | 01:52 | [273.9, 292.2] |
| 203 | 22 July | 01:47 | [272.3, 291.7] |
| 242 | 30 August | 00:15 | [274.1, 298.7] |
| 254 | 11 September | 00:41 | [278.0, 306.9] |
| 274 | 1 October | 00:17 | [279.9, 311.6] |
| 295 | 22 October | 01:37 | [285.1, 324.5] |
| 297 | 24 October | 01:25 | [280.5, 326.6] |
| 322 | 18 November | 01:20 | [276.3, 323.2] |
| 354 | 20 December | 01:22 | [289.7, 332.8] |

[1] Note: The minimum and maximum values in brackets are given for each parameter.

### 2.3. In Situ Measurements and Meteorological Data

The in situ measurements collected through the 'Risk Management through Soil Moisture Monitoring Project' in Victoria were used in this study, which refers from the Agriculture Victoria website (https://agriculture.vic.gov.au). This project, conducted by the Department of Economic Development, Jobs, Transport and Resources (DEDJTR), commenced in 2011 to provide live, profile SM data to help dryland croppers, farmers,

and advisors/managers validate the SM monitoring technology, as well as conducting training to interpret the data for crop decision making. Monitoring sites have been set up in cropping and pasture regions throughout the state by Agriculture Victoria. With the use of capacitance probes, these sites are being monitored to provide hourly measurements of soil water content and soil temperature through the soil profile. Meteorological variables such as precipitation, wind speed, and humidity have also been routinely measured at each site. The SM monitoring probes in dryland cropping areas record hourly measurements of soil water content at one source point from 30 cm down to 100 cm as a reference point for a paddock. Eight cropping monitoring sites—Werrimul, Ouyen, Speed, Bangerang, Taylors Lakes, Coonooer Bridge, Elmore, and Hamilton—are evenly distributed in western and northern Victoria. The SM monitoring probes in pasture regions provide real SM content data at depths from 10 to 80 cm with the interval of 10 cm. Some recently installed SM probes are located in the middle and southern part of Victoria including Greta, Bairnsdale, Yarram, Longwarry, Baynton Basalt, Baynton Granite, Jancourt and Lawloit.

In this study, another SM product derived from a model called the Australian Water Resource Assessment Landscape (AWRA-L) is also used. It is a daily 0.05 degree (approximately 5 km) grid-based water balance model, which provides daily, monthly and yearly estimates of surface runoff, SM, evapotranspiration and deep drainage over the whole of Australia [42]. SM estimate of the AWRA-L model contains the soil water stores in three soil layers (upper: 0–10 cm, lower: 10–100 cm, and deep: 1–6 m). As shown in Figure 2, twenty weather stations with distinctive characteristics of land cover types were selected in this study. Details of these weather stations are listed in Table 2.

**Table 2.** Details of the weather stations and corresponding land cover types.

| Station ID | Station Name | Latitude | Longitude | Elevation (m) | Land Cover |
|---|---|---|---|---|---|
| S1 | Kaniva | −36.3721°S | 141.2422°E | 142 | Croplands |
| S2 | Warracknabeal (Earlstan) | −36.2705°S | 142.2162°E | 118 | Croplands |
| S3 | Natimuk | −36.7416°S | 141.9429°E | 122 | Croplands |
| S4 | Drung Drung | −36.7768°S | 142.3937°E | 146 | Croplands |
| S5 | Warranooke (Glenorchy) | −36.7259°S | 142.7294°E | 150 | Croplands |
| S6 | Boigbeat | −35.5504°S | 142.9207°E | 60 | Grasslands |
| S7 | Birchip (Woodlands) | −35.9244°S | 142.851°E | 100 | Grasslands |
| S8 | Quambatook South | −35.9296°S | 143.4992°E | 95 | Grasslands |
| S9 | Linga | −35.1683°S | 141.6922°E | 70 | Grasslands |
| S10 | Glenalbyn (Brenanah) | −36.5486°S | 143.6958°E | 213 | Grasslands |
| S11 | Grampians (Mount William) | −37.295°S | 142.6039°E | 1150 | Forests |
| S12 | Mount Richmond | −38.1968°S | 141.3577°E | 133 | Forests |
| S13 | Benwerrin | −38.4831°S | 143.9142°E | 385 | Forests |
| S14 | Beech Forest | −38.6219°S | 143.5622°E | 443 | Forests |
| S15 | Blackwood | −37.4677°S | 144.3075°E | 547 | Forests |
| S16 | Mount Buller | −37.145°S | 146.4394°E | 1707 | Forests |
| S17 | Mount Tamboritha | −37.4667°S | 146.6883°E | 1446 | Forests |
| S18 | Mount Hotham | −36.9772°S | 147.1342°E | 1849 | Forests |
| S19 | Mount Moornapa | −37.7481°S | 147.1428°E | 480 | Forests |
| S20 | Mount Buffalo Chalet | −36.722°S | 146.8189°E | 1350 | Forests |

## 3. Methodology

### 3.1. Modified Perpendicular Drought Index (MPDI)

Richardson and Wiegand [43] described the spectral features of the NIR (near infrared)-Red spectral space, and Zhan et al. [44] found that the two-dimensional scatter plot in the spectral space shapes like a typical triangle, which can characterize quantitatively the spectral behavior of surface vegetation coverage and SM. Based on the Landsat and MODIS data, Ghulam [45] further investigated whether the distance from the soil line to any point may indicate the drought severity and SM variation. For a specific non-vegetated surface, a drought index called PDI was developed based on the scattering characteristics

of SM in the NIR-Red spectral space. However, soil heterogeneity and land surface types are not taken into account in the PDI, which might lead to the same PDI value and same perpendicular distance from a reference soil line for areas with full vegetation cover, partial cover and dry bare soil. As a consequence, Ghulam et al. [41] propose a modified PDI by introducing the fraction of vegetation into PDI to separate the effects of vegetation. The modified perpendicular drought index (MPDI) can be expressed as follows:

$$\text{MPDI} = \frac{R_{\text{Red}} + MR_{\text{SWIR}} - f_v(R_{v,\text{Red}} + MR_{v,\text{SWIR}})}{(1 - f_v)\sqrt{M^2 + 1}} \tag{1}$$

where $R_{\text{Red}}$ and $R_{\text{SWIR}}$ are the atmospherically corrected surface reflectance of red and short-wave infrared (SWIR) bands of remotely sensed data. $R_{v,\text{Red}}$ and $R_{v,\text{SWIR}}$ are regarded as the pure vegetation reflectance in the red and SWIR bands with the values of 0.05 and 0.3, respectively [41]. $M$ represents the slope of the soil line in the SWIR-Red feature space. $f_v$ is the fraction of vegetation calculated using power function of the scaled NDVI [46].

### 3.2. MPDI-NDVI Triangle Method

Numerous studies have revealed that MPDI has potential advantages for estimating SM in the areas where surface cover types vary from bare soil to densely vegetated surfaces [47,48]. It provides information about evaporation of moisture in soil and conditions of vegetation on the ground. The greater the value of MPDI is, the more arid the surface is, and vice versa [49]. As illustrated in Figure 3, MPDI typically shows a strong negative relationship with NDVI and the scatter plot seems to take a triangular shape. In the triangle, different edges can be used to represent the extreme conditions of evapotranspiration and SM. The top edge of the triangle, defined as the dry edge, represents zero evapotranspiration or low SM in the root zone and can be fitted to a negative slope using the least square method. The base edge of the triangle describes a horizontal line and corresponds to high SM condition or maximum evapotranspiration, which is called the wet edge. The group of points between the dry and wet edges indicates SM or evapotranspiration for different surface types and varying drought conditions. In the vertical direction, MPDI increases progressively from the minimum value to the maximum when the NDVI value is constant, revealing that SM decreases from maximum to minimum values correspondingly. Thus, every point in the MPDI-NDVI space strongly relates to SM availability.

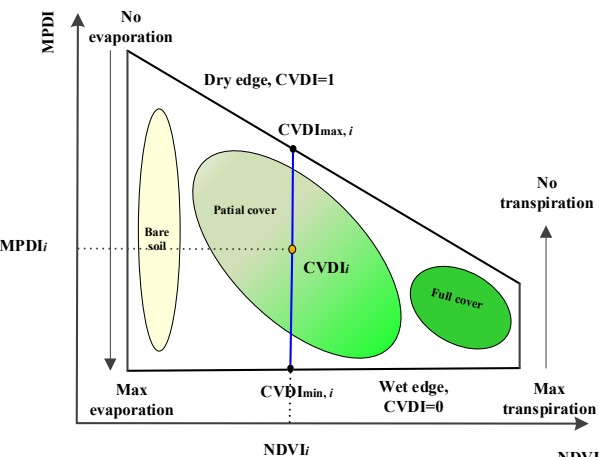

**Figure 3.** MPDI-NDVI feature space and definition of the condition vegetation drought index (CVDI) (reproduced from Sandholt et al. [19]).

Similar shape and spectral patterns are obtained in LST-NDVI space and Sandholt et al. [19] put forward the fundamental theory and concept of temperature vegetation dryness index (TVDI) based on the LST-NDVI space. In the present paper, an MPDI-NDVI

index (condition vegetation drought index, CVDI) was proposed by substituting the LST in the LST-NDVI space with MPDI, which is similar to TVDI mathematically.

$$CVDI = \frac{MPDI - MPDI_{\min}}{MPDI_{\max} - MPDI_{\min}} \qquad (2)$$

where $MPDI_{\max}$ and $MPDI_{\min}$ are the maximum and minimum of MPDI with the same surface cover type on the dry and wet edges, which can be defined as follows:

$$MPDI_{\min} = a_1 \cdot NDVI + b_1 \qquad (3)$$

$$MPDI_{\max} = a_2 \cdot NDVI + b_2 \qquad (4)$$

where $a_1$, $b_1$, $a_2$ and $b_2$ are the fitting coefficients on the dry and wet edges.

The flowchart of constructing the CVDI in this study is presented in Figure 4. The main advantages of MPDI-NDVI are (1) higher sensitivity of drought indices to SM variations over vegetated areas, (2) it is easier to monitor SM and drought without any ancillary data, (3) absolute higher accuracy in MPDI retrieval compared with the LST retrieval in the LST-NDVI space, (4) it could eliminate the influence of terrain and latitude variations effectively. Limitations of this method also lie on the requirement of a large range of regions with all land cover types from bare land to fully vegetated cover and the subjective in boundary fitting between the dry and wet edges of a specific feature space.

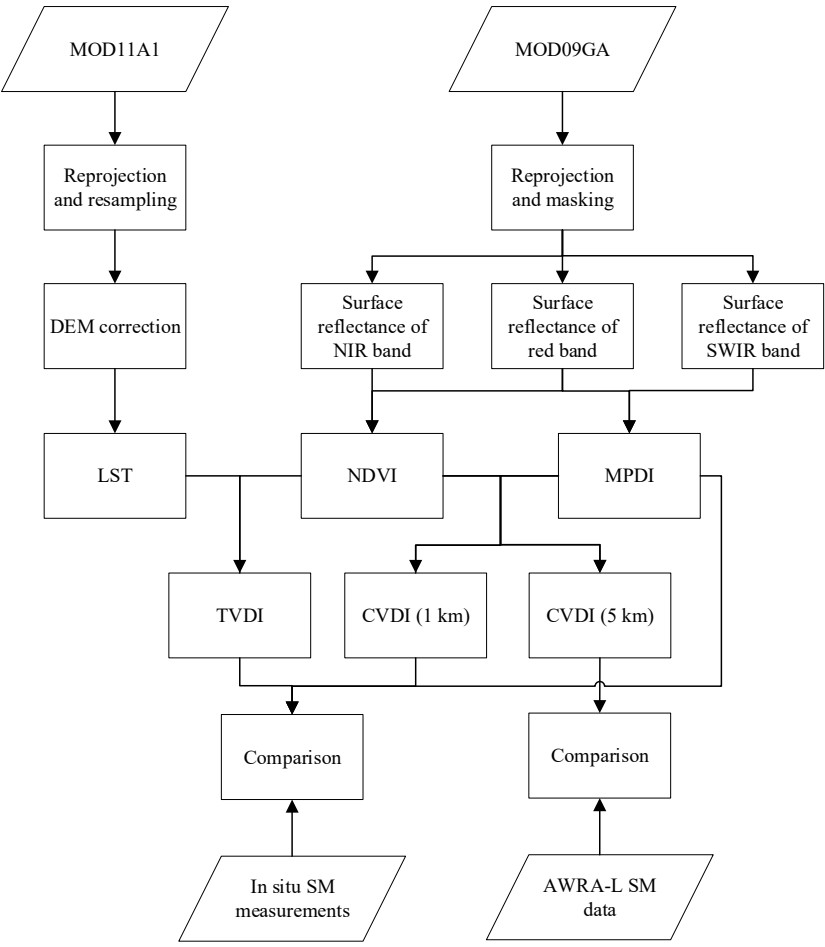

**Figure 4.** Flowchart of constructing CVDI and soil moisture (SM) monitoring in this study.

## 4. Results

### 4.1. Estimation of the Dry and Wet Edges for TVDI and CVDI

The scatterplots of NDVI pixels against corresponding MPDI and LST from DOY004 to DOY354 are shown in Figure 5. An iterative process proposed by Tang et al. [50] was used to determine the dry and wet edges in the MPDI/LST-NDVI feature space. From Figure 5, it is obvious that the frames of dry and wet edges for TVDI (red lines) and CVDI (blue lines) presented on these days are close to the conceptual triangle or trapezoid space indicated in Figure 3. All dry edges clearly demonstrate the negative relationship between MPDI/LST and NDVI and wet edges are almost constant and consistent in the two spaces. In addition, some issues might need to be explained and discussed in the two presented spaces.

(1) For all days, the upper blue lines representing dry edges of MPDI-NDVI space indicated significant fitted linear regressions between MPDI and NDVI. The coefficients of determination ($R^2$) produced by the CVDI method were significantly higher than the corresponding $R^2$ values calculated by the TVDI method. The result reveals that the MPDI-NDVI space was clearly triangular or trapezoid with a distinct division of dry and wet edges. In comparison with TVDI, the CVDI method is more applicable and feasible in the study area.

(2) A phenomenon of "tail down" structure frequently occurs on the upper red lines indicating dry edges of LST-NDVI space with the NDVI values in the range of 0–0.2, which corresponds to land surface types covered by bare soil or sparse vegetation in remote sensing images. It will cause the simulated LSTmax on dry edges to be lower than the actual value. Although LST is sensitive to water stress under different land cover types, the dry edges could not represent the real maximum water stress. Moreover, the framework seems like a biparabolic shape, which was consistent with the research conducted by Liu et al. [32,33]. The same phenomenon happens on the dry edges of MPDI-NDVI space on individual days such as day of year (DOY) 055, 061 and 295. In general, the dry edges in MPDI-NDVI space have noticeable negative slopes and MPDImax tend to decrease with increasing of NDVI. Furthermore, MPDImax is almost larger than 1 with the NDVI values in the range of 0–0.2 on individual days, which is particularly significant on DOY004, 023, 075 and 102. It can be explained that rainfall in 2019 was less than average with January and April being particularly dry in large parts of Victoria reported by the Bureau of Meteorology (BoM) in Australia (http://www.bom.gov.au/climate/current/annual/vic/summary.shtml).

(3) As illustrated in Table 1 and Figure 5, from DOY 004 to 354, the maximum values of LST fluctuate gradually with the changes of seasons. However, as had been reported, the hottest on record in Victoria contributed to significant bushfire activity during February and March 2019. Meanwhile, the 2019–2020 Australian bushfire season, colloquially known as the Black Summer, occurred across eastern and south-eastern Victoria from November 2019. Therefore, LST values on DOY055, 061, 075, 322 and 354 might be affected by the known bushfires and overestimated on the whole for these cases. Moreover, the maximum values of NDVI from DOY004 to 354 show a slight trend and reach to the largest on DOY176 and 178. As shown in Figure 2, the largest proportion of woody vegetation is mainly distributed in the eastern area called Victorian Alps with a relative high altitude. Considering the influence of topography and land cover types on the LST, many studies performed relative comprehensive investigations [24,26–29]. Nevertheless, bushfires and topography will not need to be responsible for the estimation of MPDI.

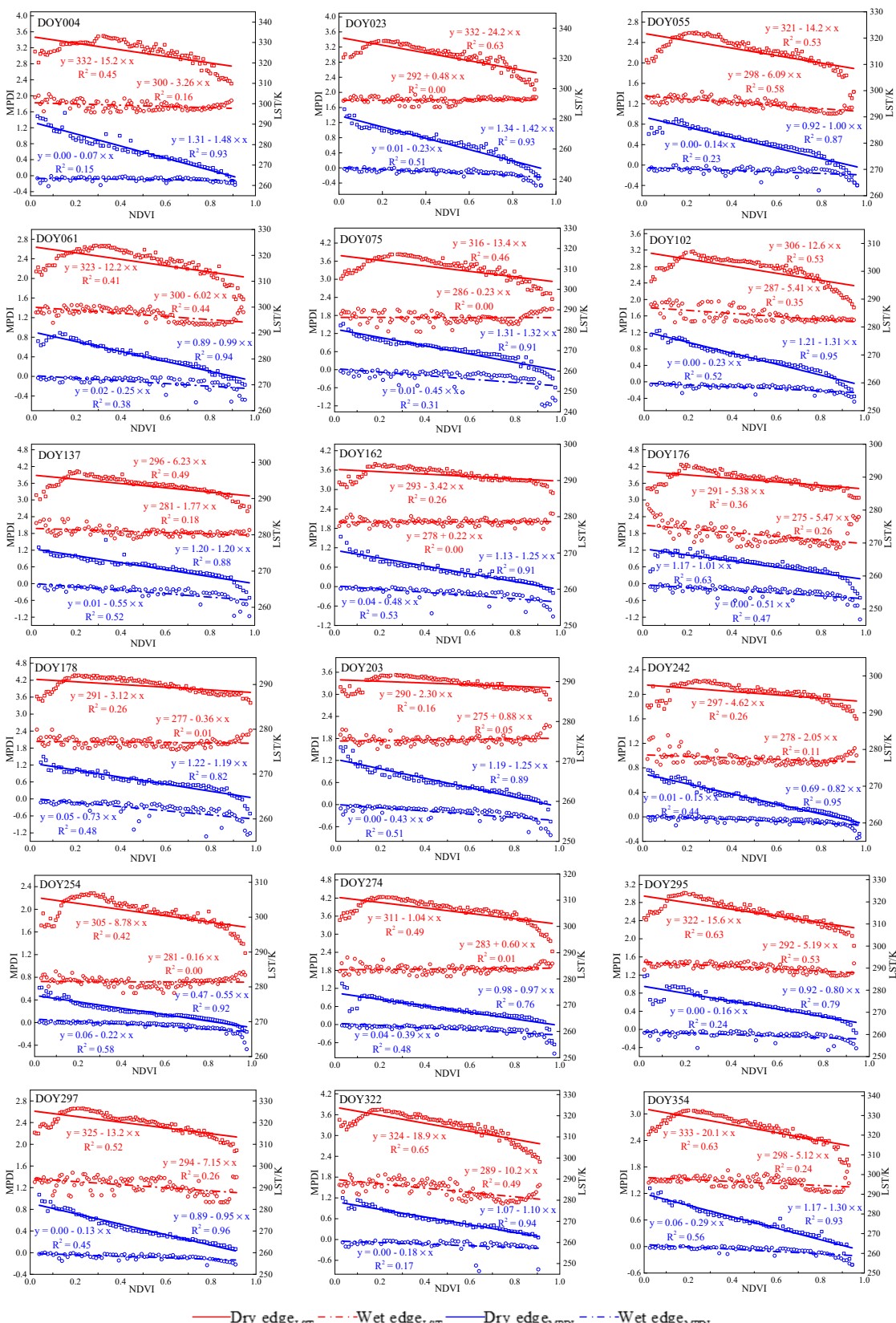

**Figure 5.** Scatterplots of dry and wet edges in the MPDI-NDVI and land surface temperature (LST)-NDVI feature spaces for 18 images.

*4.2. Spatial Comparison of SM Estimation between MPDI, TVDI and CVDI*

Based on Equations (1) and (2) and the constraints of LST/MPDI-NDVI feature spaces in Figure 5, the MPDI, TVDI and CVDI were retrieved with the daytime images for 18 days in the year 2019. In order to compare the feasibility and applicability of the three drought indices in SM estimation, station observations at crop sites in Victoria were used and the value difference between SM measurements and drought indices was evaluated using the $R^2$ and root mean square error (RMSE). Considering SM measured in the depth of 30–100 cm at crop sites, we chose the ground measurements at the 30 cm of the soil profile for the model evaluation in this study. Table 3 shows the evaluation results between drought indices and SM for all of 18 days and 0.01 and 0.05 confidence levels. As can be seen from the CVDI, poor correlation is found on DOY 176 and 203 with the $R^2$ of less than 0.12. The TVDI and MPDI also have a little poor correlation with SM at the two days probably because of some outliers related with cloud cover or some missing pixels of LST. That is to say, some particular pixels are not consistent with the general distribution pattern of image values and the corresponding indices are misestimated compared with their surrounding pixels. The CVDI performs well with the $R^2$ of more than 0.4 on DOY004, 023, 055, 061, 102, 137, 242, 274, 295 and 322, which are significantly higher than the corresponding $R^2$ values produced by TVDI and MPDI except for DOY055, 061 and 242. In general, judging from the values of the $R^2$ and RMSE, CVDI provides an apparently more accurate capture of the spatial variation in SM than TVDI and MPDI. Furthermore, CVDI and MPDI perform better in summer than that in winter from DOY162 to 254 due to more rain in winter at the study site, while the estimated result of the TVDI does not show great seasonal difference because the dense evergreen trees and high-altitude mountains in the northwest of Victoria have a great influence on the TVDI estimation.

**Table 3.** Evaluation results between drought indices and SM for all of 18 days.

| DOY | CVDI | | TVDI | | MPDI | |
|---|---|---|---|---|---|---|
| | $R^2$ | RMSE | $R^2$ | RMSE | $R^2$ | RMSE |
| 004 | 0.48 | 0.05 | 0.31 | 0.09 | 0.21 | 0.10 |
| 023 | 0.67 * | 0.04 | 0.42 | 0.10 | 0.57 * | 0.09 |
| 055 | 0.54 * | 0.05 | 0.57 * | 0.05 | 0.48 | 0.06 |
| 061 | 0.45 | 0.07 | 0.48 | 0.06 | 0.41 | 0.07 |
| 075 | 0.33 | 0.06 | 0.11 | 0.06 | 0.29 | 0.09 |
| 102 | 0.47 | 0.04 | 0.14 | 0.14 | 0.47 | 0.06 |
| 137 | 0.67 * | 0.06 | 0.51 | 0.10 | 0.64* | 0.09 |
| 162 | 0.26 | 0.08 | 0.38 | 0.10 | 0.37 | 0.12 |
| 176 | 0.12 | 0.08 | 0.15 | 0.10 | 0.10 | 0.22 |
| 178 | 0.39 | 0.08 | 0.33 | 0.09 | 0.25 | 0.18 |
| 203 | 0.06 | 0.06 | 0.11 | 0.12 | 0.12 | 0.21 |
| 242 | 0.42 | 0.09 | 0.44 | 0.10 | 0.17 | 0.11 |
| 254 | 0.28 | 0.07 | 0.55 * | 0.12 | 0.13 | 0.09 |
| 274 | 0.67 * | 0.08 | 0.61 * | 0.13 | 0.49 | 0.18 |
| 295 | 0.45 | 0.09 | 0.31 | 0.17 | 0.40 | 0.19 |
| 297 | 0.35 | 0.07 | 0.14 | 0.16 | 0.30 | 0.20 |
| 322 | 0.71 ** | 0.07 | 0.04 | 0.12 | 0.16 | 0.23 |
| 354 | 0.24 | 0.07 | 0.24 | 0.07 | 0.01 | 0.16 |

Relationships significant at *P* < 0.05 (*) and *P* < 0.01 (**) are shown.

In order to further conduct the spatial and temporal variations of MPDI, TVDI and CVDI, the spatial distributions of the three drought indices retrieved on DOY004, 242 over the whole study area are shown in Figure 6. As shown in the figure, similar spatial patterns in the distribution of the three drought indices are observed except for the MPDI and CVDI on DOY242. Low values of the three indices on DOY004 and the TVDI on DOY242 in the east are found where a large proportion of dense woody vegetation in Victoria is distributed. That means there is an apparently higher soil water availability in the east

than those in other parts. It should also be mentioned that, for the TVDI in the two days, the savannas and shrublands located in northwest Victoria in Figure 2 seem to have higher TVDI values, which is an opposite phenomenon to the CVDI and MPDI over the plain areas. The values of CVDI and MPDI on DOY242 both have a relatively small difference between the mountain and plain areas, which indicates that the plain areas in Victoria are mostly covered with crop and grass in winter. However, the TVDI on DOY242 has no similar spatial pattern because the LST is greatly affected by the topography and DEM data is considered to calibrate the LST in the mountain areas [24,26].

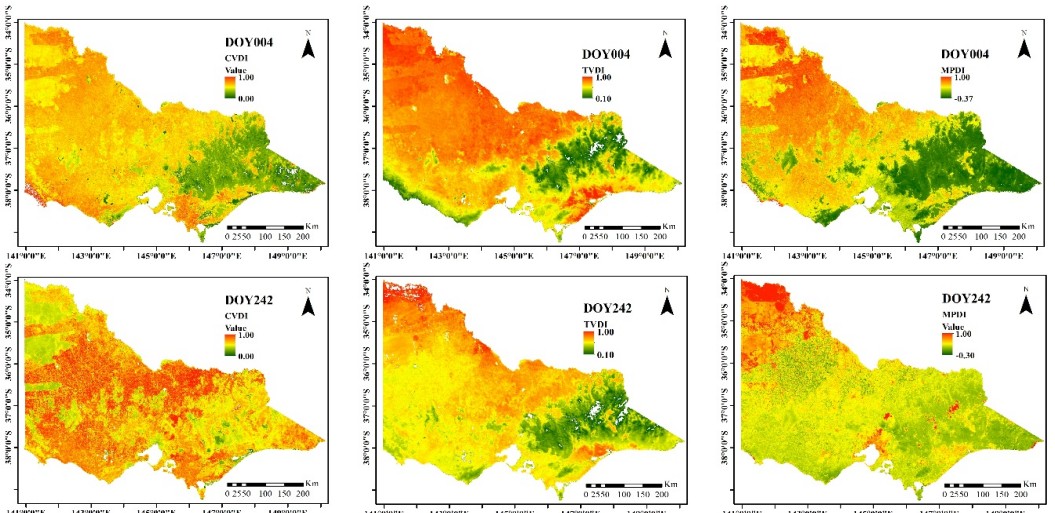

**Figure 6.** Spatial distribution of the three drought indices on DOY004 and 242.

Both the TVDI and CVDI were retrieved under the constraint of dry and wet edges in this study. LST and MPDI adopted to represent water stress between canopy and transpiration in the feature spaces of TVDI and CVDI, respectively. Consequently, in the parameterization scheme of CVDI, there will be an inseparable relationship between MPDI and CVDI. In order to further investigate the relationship between MPDI, TVDI and CVDI, scatterplots between the three indices obtained at the crop sites on DOY004, 242 are demonstrated in Figure 7. Better correlation relationships between MPDI and CVDI are presented than those between TVDI and CVDI in both two days with the difference in $R^2$ of 0.25 on DOY004 and 0.15 on DOY242. This can partly explain the similar spatial pattern in the distribution of MPDI and CVDI.

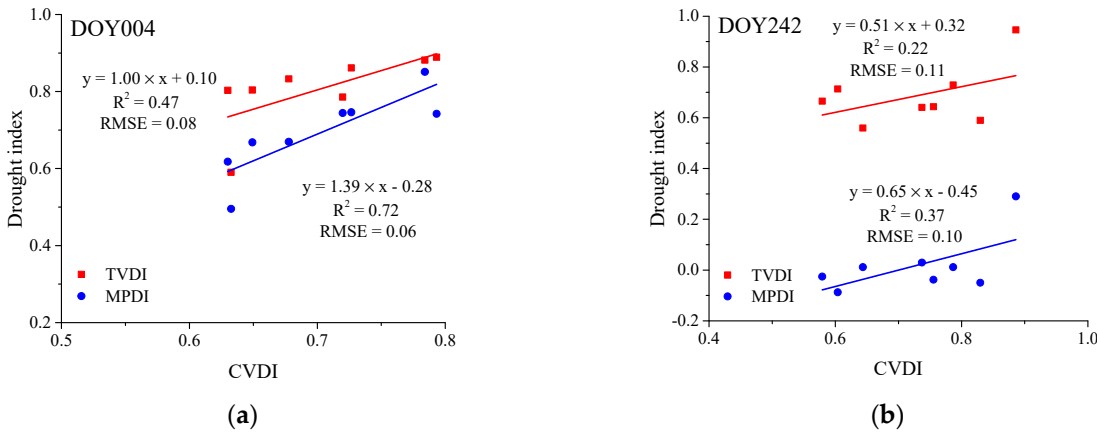

**Figure 7.** Scatterplots between the three drought indices obtained at the crop sites on DOY004 (**a**), 242 (**b**).

*4.3. Temporal Comparison between MPDI, TVDI and CVDI Based on AWRA-L SM Data*

In this work, the upper AWRA-L SM data are used to compare with CVDI and ground SM measurements at pasture sites. In order to investigate sensitivity between the CVDI values and AWRA-L SM data, the CVDI retrieved at a 5 km resolution was estimated based on the MODIS data. Table 4 shows the evaluation results between the CVDI values at 1 km (*x*-axis) and 5 km (*y*-axis) resolutions as well as their corresponding value range for all of 18 days and 0.01 and 0.05 confidence levels. The $R^2$ between $CVDI_{1 km}$ and $CVDI_{5 km}$ ranges from 0.26 to 0.75 (P < 0.01), while the RMSE ranges from 0.07 to 0.18. The correlation relationship is acceptable on most days considering the complex land cover, except for DOY137, 178 and 203. Table 4 also presents the estimation results between the $CVDI_{5 km}$ and AWRA-L SM data for the 18 days. Judging from the values of $R^2$, a poor correlation relationship is observed on DOY23, 75, 162, 254 and 322 where the $R^2$ achieved is lower than 0.1, while high values of $R^2$ are found on DOY137, 176, 203, 274 and 297.

**Table 4.** Estimation results between $CVDI_{5 km}$ and $CVDI_{1 km}$, Australian Water Resource Assessment Landscape (AWRA-L) SM data for the 18 days.

| DOY | $R^2$ between $CVDI_{1 km}$ and $CVDI_{5 km}$ | RMSE between $CVDI_{1 km}$ and $CVDI_{5 km}$ | AWRA-L SM Data | |
|---|---|---|---|---|
| | | | $R^2$ | RMSE |
| 004 | 0.75 ** | 0.07 | 0.14 | 0.14 |
| 023 | 0.73 ** | 0.08 | 0.10 | 0.15 |
| 055 | 0.62 * | 0.12 | 0.20 | 0.17 |
| 061 | 0.49 | 0.10 | 0.24 | 0.12 |
| 075 | 0.58 * | 0.07 | 0.07 | 0.10 |
| 102 | 0.36 | 0.09 | 0.22 | 0.09 |
| 137 | 0.29 | 0.15 | 0.44 | 0.13 |
| 162 | 0.45 | 0.11 | 0.01 | 0.15 |
| 176 | 0.66 * | 0.10 | 0.79 ** | 0.08 |
| 178 | 0.26 | 0.12 | 0.28 | 0.11 |
| 203 | 0.28 | 0.13 | 0.53 * | 0.11 |
| 242 | 0.71 ** | 0.09 | 0.26 | 0.15 |
| 254 | 0.54 * | 0.10 | 0.09 | 0.14 |
| 274 | 0.63 * | 0.13 | 0.53 * | 0.15 |
| 295 | 0.61 * | 0.15 | 0.09 | 0.23 |
| 297 | 0.36 | 0.18 | 0.89 ** | 0.08 |
| 322 | 0.48 | 0.11 | 0.03 | 0.15 |
| 354 | 0.52* | 0.13 | 0.13 | 0.17 |

Relationships significant at *P* < 0.05 (*) and *P* < 0.01 (**) are shown.

Figure 8 shows the temporal variations of ground measurements, AWRA-L SM data, and the CVDI at 1 and 5 km resolutions with the monthly cumulative precipitation for all of the eight sites. The ground measurements and the AWRA-L SM data increased during the high values of precipitation and decreased before and after rainfall events. However, the AWRA-L SM data perform abnormally on DOY176, 178 and 203 in almost all of the sites with relatively low values. Correspondingly, $CVDI_{1 km}$ and $CVDI_{5 km}$ have very high values at the three days indicating that the land is extremely dry during that period. It can be explained from the two aspects. First, the monthly precipitation is a cumulative value and there is no rainfall before and during that period. Second, vegetation growth leads to a water shortage of the upper surface, especially in winter in Australia. Compared with ground measurements, the AWRA-L SM data are apparently lower but have a wider value range, which can be found in all of the eight sites. Meanwhile, the AWRA-L SM data and CVDI have similar trends. Thus, at pasture sites, CVDI provides a better representative of the spatial and temporal variations in AWRA-L SM data than that of ground SM measurements.

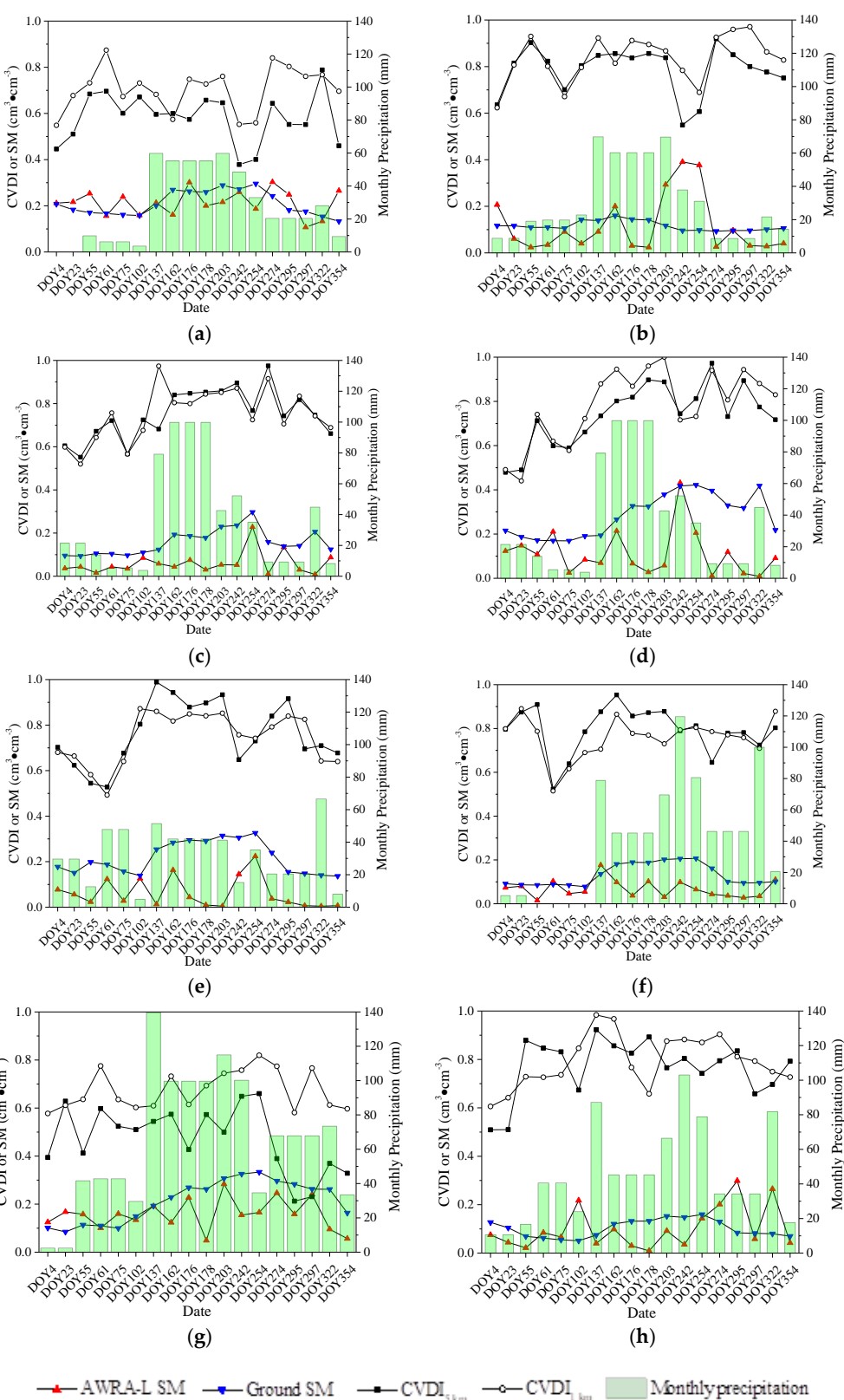

**Figure 8.** Time series of the AWRA-L SM data, ground measurements, and the CVDI at 1 and 5 km resolutions with the monthly cumulative precipitation for 8 sites including (**a**) Lawloit, (**b**) Greta, (**c**) Baynton Basalt, (**d**) Baynton granite, (**e**) Bairnsdale, (**f**) Longwarry, (**g**) Jancourt and (**h**) Yarram.

In order to further explore the temporal variations of the MPDI, TVDI and CVDI, the AWRA-L SM data presented in Table 2 are used in this work. Table 5 shows the $R^2$ and RMSE between the indices and AWRA-L SM data and 0.001, 0.01 and 0.05 confidence levels. The results indicate that the CVDI produces the best performance as for the comparison with the TVDI and MPDI. A good correlation relationship between CVDI and SM data is found in most of the 20 stations where the $R^2$ achieved can reach to 0.69 ($P < 0.001$). The results produced by MPDI and CVDI are generally similar to each other at croplands and grasslands, while the results are achieved by the MPDI with the $R^2$ relatively lower than that obtained by the CVDI at forests. TVDI performs poorly and the $R^2$ is lower than that achieved by the CVDI and MPDI for almost all the stations. Judging from the RMSE, CVDI also produces the lowest RMSEs at every land cover type, while MPDI has very high values of RMSEs, especially at forests. It can be concluded that CVDI can accurately capture the seasonal variations of SM for different land cover and be used effectively for SM and drought monitoring.

**Table 5.** Evaluation results between drought indices and AWRA-L SM data for all of the 20 stations.

| Station ID | CVDI | | TVDI | | MPDI | |
|---|---|---|---|---|---|---|
| | $R^2$ | RMSE | $R^2$ | RMSE | $R^2$ | RMSE |
| S1 | 0.14 | 0.04 | 0.34 * | 0.10 | 0.39 ** | 0.09 |
| S2 | 0.41 ** | 0.05 | 0.29 * | 0.11 | 0.35 * | 0.13 |
| S3 | 0.55 *** | 0.06 | 0.20 | 0.11 | 0.40 * | 0.10 |
| S4 | 0.69 *** | 0.05 | 0.21 | 0.10 | 0.51 *** | 0.11 |
| S5 | 0.53 *** | 0.09 | 0.19 | 0.11 | 0.52 *** | 0.14 |
| S6 | 0.67 *** | 0.06 | 0.16 | 0.09 | 0.49 *** | 0.11 |
| S7 | 0.50 *** | 0.04 | 0.48 *** | 0.06 | 0.52 *** | 0.08 |
| S8 | 0.56 *** | 0.05 | 0.18 | 0.09 | 0.55 *** | 0.08 |
| S9 | 0.40 ** | 0.04 | 0.06 | 0.08 | 0.46 ** | 0.09 |
| S10 | 0.11 | 0.07 | 0.18 | 0.11 | 0.09 | 0.12 |
| S11 | 0.13 | 0.10 | 0.11 | 0.13 | 0.24 * | 0.35 |
| S12 | 0.48 *** | 0.15 | 0.00 | 0.20 | 0.15 | 0.35 |
| S13 | 0.21 | 0.14 | 0.04 | 0.18 | 0.20 | 0.17 |
| S14 | 0.58 *** | 0.18 | 0.02 | 0.14 | 0.19 | 0.24 |
| S15 | 0.30 * | 0.07 | 0.14 | 0.10 | 0.28 * | 0.10 |
| S16 | 0.27 * | 0.15 | 0.00 | 0.13 | 0.11 | 0.20 |
| S17 | 0.38 ** | 0.08 | 0.34 * | 0.11 | 0.46 ** | 0.11 |
| S18 | 0.00 | 0.14 | 0.09 | 0.14 | 0.04 | 0.35 |
| S19 | 0.28 * | 0.14 | 0.04 | 0.13 | 0.13 | 0.19 |
| S20 | 0.27 * | 0.10 | 0.00 | 0.11 | 0.12 | 0.12 |

Relationships significant at $P < 0.05$ (*), $P < 0.01$ (**) and $P < 0.001$ (***) are shown.

## 5. Discussion

Compared with the parameterization scheme of MPDI and TVDI, CVDI combines the significant information from MPDI and NDVI and establishes the SM-estimated mechanisms just as TVDI does. MPDI describes the SM status by operating the vector parallel to the soil line, which cannot be determined effectively and accurately. The states of vegetation such as greenness, stress levels, and activity rates significantly lead to the different performance of MPDI for SM estimation over sparsely vegetated and densely vegetated surfaces [41]. TVDI provides complete information on SM based on the LST values and given vegetation index derived from satellite data. The variations in LST are usually affected by the air and canopy, which causes TVDI to vary with terrain and latitude, especially in mountainous areas [30]. In this study, MPDI was used as an alternative to LST in the establishment of the CVDI model, which indicated that CVDI could depict the SM status and vegetation growth precisely from bare soil to densely vegetation-covered surfaces. Meanwhile, CVDI can eliminate the influence of terrain and latitude variation on LST and is more sensitive to SM change by substituting LST with MPDI.

In Section 4.2, CVDI shows a more favorable correlation with measured SM than MPDI and TVDI, there are still individual cases where the $R^2$ is rather low, especially on DOY102, 297 in Table 3. Several factors may contribute to these errors and uncertainties.

### 5.1. The Effect of Land Cover Types on Three Drought Indices

As shown in the land cover map produced from the MODIS image (Figure 2), the land cover types related to the vegetation-covered surfaces were classified into four categories, namely, cropland, forests, grassland and shrubs. Cropland is a land cultivated with crops including cereals, oilseeds and grain that are located at low elevations and plain areas. Forests are mostly located in the eastern part of the study area and comprise of many kinds of forests such as evergreen needleleaf forests, evergreen broadleaf forests and deciduous broadleaf forests at relatively higher elevations. Grassland is characterized by low vegetation covers and comprises annual and perennial short grasses located in the plain areas. Shrubs are mainly distributed in the north-western part called Big Desert Wilderness Area and vegetation cover compositions are entirely different from cropland, forests and grassland, which consist of open and closed shrubs.

Scatterplots established between the average values of MPDI, the TVDI, the CVDI and SM in the four main land cover classes are presented in Figure 9. The MPDI values are lower in the forest region, whereas cropland, grassland and shrubs exhibit higher MPDI values and showing significant seasonal dynamics. No correlation between the MPDI and SM in forests indicates that MPDI is less sensitive to water availability in the forest region. TVDI and CVDI generally present higher values than MPDI and show the similar trends in the four land cover regions, while TVDI values were found to be higher than CVDI, especially in grassland and shrubs. Although the trend of the TVDI and CVDI is similar to that of SM in any of the four land cover types, closer agreements with SM for the CVDI are presented. In the cropland and grassland, the CVDI values fluctuate considerably with the changes of SM, while no significant fluctuations are reflected in the TVDI values. It can be concluded that the CVDI generally display the water stress with the growing season and the changes of CVDI values are closely dependent on the water availability and influenced by the four land cover types.

It also can be seen from Figure 8 that the monthly precipitation reveals the discipline of total precipitation during the whole year in the study area: more precipitation in winter and less in summer. Similar trends between the CVDI, ground SM measurements and monthly precipitation were found indicating that precipitation is also a key factor influencing the water availability and changes of the drought indices. It can be inferred that the discipline of monthly precipitation can be applied in the four land cover regions in Figure 9. Thus, it can be concluded that the TVDI shows less sensitivity to the changes of precipitation, especially in winter. The MPDI and monthly precipitation have no significant relationship in the forest region, while good relationships were observed between the CVDI and precipitation in all land cover types. Therefore, it shows that the CVDI responds significantly to the recent precipitation and provides information on the seasonal dynamics of water status.

### 5.2. The Effect of Field Measurements and Remote Sensing Data

As seen in Figure 6, many missing pixels were found in the spatial distribution of TVDI on both DOY004 and 242. A further investigation presents that the gap information is related with the missing values of MODIS LST data. The quality of MODIS LST product is mainly responsible for such gap. Same phenomenon also occurs in the distribution map of the CVDI on DOY242. This can be explained by the fact that NDVI image rather than the LST data relatively affects the CVDI estimation: the NDVI value of water body is less than 0, which should be eliminated when constructing MPDI-NDVI space. It should be noted that the CVDI does not show a more favorable performance than the TVDI and MPDI in Table 3. It might be explained from two aspects. First, in this work, in situ SM measurements on the point scale were used to evaluate the effectiveness of the CVDI. The CVDI pixel value calculated from the remote sensing data is inconsistent with the measured SM because of

the mismatch in spatial scale [36]. Consequently, the CVDI could not absolutely represent the volumetric SM measured by the ground instrumentation [51]. Second, there are not enough measurements both at crop and pasture sites in local stations. The reliability and applicability of the CVDI should be further evaluated by measuring more observed data in different land cover regions. Besides, the CVDI model was estimated to monitor SM only on the daily scale in this study. The biweekly, monthly and yearly scales are often used in similar studies, and better performances were generally observed in the SM and drought monitoring [22,30,31,52]. Thus, a further study is still needed on these temporal scales. Additionally, as seen in Figure 9, for measured SM the in cropland region, the influence of irrigation should also be considered in the model evaluation. Reasonable irrigation can affect the field water holding status and crop growth, which is of great importance for remote-sensing-based drought indices monitoring.

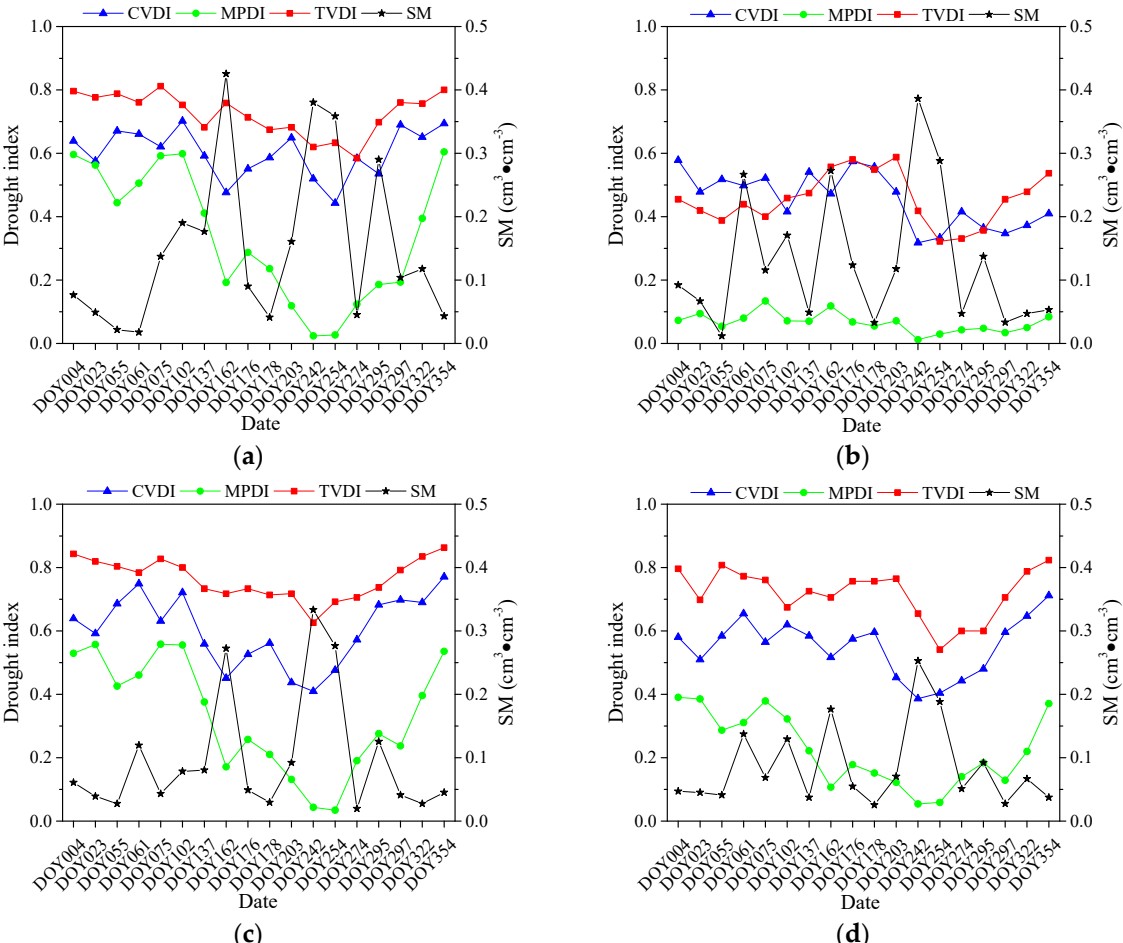

**Figure 9.** Scatterplots established between the three drought indices and SM, monthly precipitation in the four main land cover types including (**a**) Cropland, (**b**) Forest, (**c**) Grassland and (**d**) Shrubs.

## 6. Conclusions

The temperature vegetation dryness index based on LST-NDVI space has been widely applied to monitor SM and drought on a regional scale from remotely sensed data. Considering the effect of terrain variations on this method and mismatch of spatial scale between visible, infrared and thermal bands data, a new drought index called the condition vegetation drought index (CVDI) was firstly proposed to explore the potential of it for SM estimates in the present paper. MPDI was used to substitute for the LST and construct a new feature space (MPDI-NDVI trapezoidal space) with quite similar shape and spectral patterns. The proposed index was demonstrated based on time series of MODIS data for

18 days and in situ SM measurements over a large range of regions at crop and pasture sites in Victoria, Australia. Some conclusions are listed as follows:

(1) The framework of the LST-NDVI space does not seem like a designed trapezoid shape, and the LST on the dry edge often underestimates the real maximum water stress. In the parameterization scheme of CVDI, the dry and wet edges in the MPDI-NDVI space form a more significant trapezoid shape for all of the 18 days. Higher correlation relationships are generally observed on the dry edge in the MPDI-NDVI space. In addition, the CVDI will not be greatly affected by the terrain and altitude variations.

(2) Comparison with the MPDI, TVDI for SM estimation at crop sites was demonstrated to evaluate the effectiveness of the CVDI for 18 days. The results indicate that the retrieved CVDI had significantly higher $R^2$ values than the MPDI and TVDI in most days and provided a more effective capture for the spatial variation of SM. Compared with the spatial distribution of the MPDI and TVDI on DOY004, 242, the CVDI showed a similar pattern with the MPDI and had a relatively small difference between the mountain and plain areas on DOY242 due to the densely vegetated cover for the two areas in winter. Meanwhile, higher correlation relationships were found between the CVDI and MPDI with the $R^2$ of 0.72 on DOY004 and 0.37 on DOY242, respectively.

(3) The sensitivity between the CVDI and AWRA-L SM products was investigated at pasture sites. The results show that the CVDI had a good negative correlation relationship with AWRA-L data on DOY137, 176, 203, 274 and 297. The comparison between ground SM measurements, AWRA-L data, CVDI and monthly cumulative precipitation for temporal variations at eight sites was also conducted. The AWRA-L data and ground measurements fluctuated significantly with temporal variations of precipitation for all of eight sites. Apparently, lower values were observed in the AWRA-L SM product but the data had a wider value range. Meanwhile, the CVDI had a closer trend with the AWRA-L data and better representatives of the spatial and temporal variation in AWRA-L data than that of MPDI and TVDI at different land cover types.

Therefore, the CVDI substantially captures the spatial-temporal variations of SM with reasonable accuracy. However, numerous confounding factors may affect the applicability of the CVDI in accurate estimates of SM. Thus, further work needs to be carried out to investigate and validate the CVDI in other different regions on the site scale. Moreover, additional work is also required to explore and optimize the parameterization scheme of the CVDI for short-term and small spatial-scale SM monitoring.

**Author Contributions:** Conceptualization, L.T. and D.R.; methodology, L.T.; software, L.T.; validation, L.T. and D.B.; data curation, D.B.; writing—original draft preparation, L.T.; writing—review and editing, D.R. and A.W.; funding acquisition, L.T. All authors have read and agreed to the published version of the manuscript.

**Funding:** This work has been supported by Chinese Natural Science Foundation Project (41901278) and supported in part by the University of Melbourne Research Contract #300405 sponsored by the Korea Aerospace Research Institute, and by the State Scholarship Fund of China.

**Institutional Review Board Statement:** Not applicable.

**Informed Consent Statement:** Not applicable.

**Data Availability Statement:** Data sharing not applicable.

**Acknowledgments:** We thank Agriculture Victoria for providing the valuable ground SM observations and technical support.

**Conflicts of Interest:** The authors declare no conflict of interest.

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
