# Peer review of "A New Drought Index for Soil Moisture Monitoring Based on MPDI-NDVI Trapezoid Space Using MODIS Data"

_remotesensing, doi:10.3390/rs13010122_

Round 1

Reviewer 1 Report

December 17, 2020

Manuscript: ‘A new drought index for soil moisture monitoring based on MPDI-NDVI trapezoid space using MODIS data’

This paper proposes a new drought index named the condition vegetation drought index (CVDI) to monitor the  temporal  and  spatial  variations  of  soil  moisture  status  by  substituting  the  land  surface  temperature (LST) with the modified perpendicular drought index (MPDI). In situ soil moisture observations at crop and pasture sites in Victoria (Australia) were used to validate the effectiveness of the CVDI. Compared with the MPDI and TVDI, results showed the superiority of the CVDI in terms of coefficient of determination (R2) and root mean square error (RMSE) when the Australian Water Resource Assessment Landscape (AWRA-L) soil moisture product was used as a benchmark data. This is a relevant topic lies within the scope of the MDPI remote sensing journal. The article is well organized and neatly written with the appropriate scientific content. Based on the above, I support the publication of this manuscript, but only after a minor revision.

********************************

Title: it fits perfectly the paper content. 

Abstract: it is quite adjusted to the paper content.

Lines 24-29: for clarity, the authors should add some statistical metrics to support these results (e.g. R2 and RMSE)   

Introduction: it provides sufficient background and includes relevant references on the features of LST-VI space and its relation with the soil moisture (SM), highlighting the main limitations of the temperature vegetation dryness index (TVDI). Objectives and the novelty are clearly presented.

Study area and Data: the study area and data sets are effectively described.

Methodology: the methodology is clearly described.

Results: these are clearly presented.

Line 313, Figure 5: I think authors should increase the size of the numbers and letters, because it is very difficult to discern results for the MPDI/LST-NDVI feature space. 

Line 325, Table 3; Line 365, Table 4; Line 393, Table 5: for clarity, the authors should highlight those values of R2 with statistical significance at 95% level.

Discussion: this is clearly presented and is supported by results from the previous section.  

Line 435, Figure 9: for clarity, I think the authors should apply a different color to each drought index or SM

Conclusion: these are clear, concise, and are in line with results.

Reviewer 2 Report

The article proposes to test a new drought index for monitoring soil moisture. Despite the interest to consider MODIS satellite data and proposed algorithm to meet this objective, some aspects need to be clarified.

  • Concerning the use of MODIS data, the database seems very limited to establish reliable statistics, why limit this analysis to 2019 and not have an analysis of all the data during 20 years of MODIS proiducts?
  • We do not see a seasonal context of the proposed index ? We know that the behaviors are not the same during the different seasons of the surface measurements (temperature, soil moisture, vegetation cover..)?
  • The moisture measurement considered is at 30cm, what effect and limitation of this choice for considering different types of land use with different root depths?
  • Why the choice of linear relations in equations (3) and (4).
  • In introduction, it is written that SAR data are proposed with a repeatability of 16-25 days, the references are relatively old, we are now with offers of 6 days with Sentinel-1 (https://doi.org/10.3390/s17091966, etc….).
  • As an introduction, I think it would be important to have a state of the art of the some important studies using soil moisture or NDVI based indices (https://doi.org/10.1080/01431169608949106, https://doi.org/10.1038/s41598-018-37911-x etc)
  • Figures (5) are not very readable
  • Figure 7, to be specified MDPI at the ordinate axis?
  • In figure 9, , it might be better to consider two ordinate axes, one for soil moisture and one for indices?

Round 2

Reviewer 2 Report

Authors consider comments and different corrections. I propose acceptance of the proposed paper.